# High-Level Path Planning for an Autonomous Sailboat Robot Using Q-Learning

**DOI:** 10.3390/s20061550

**Published:** 2020-03-11

**Authors:** Andouglas Gonçalves da Silva Junior, Davi Henrique dos Santos, Alvaro Pinto Fernandes de Negreiros, João Moreno Vilas Boas de Souza Silva, Luiz Marcos Garcia Gonçalves

**Affiliations:** 1Universidade Federal do Rio Grande do Norte, DCA-CT-UFRN, Campus Universitario, Lagoa Nova, Natal, RN 59078-970, Brazildavihenriqueds@gmail.com (D.H.d.S.); alvarodenegreiros@gmail.com (A.P.F.d.N.); 2Instituto Federal do Rio Grande do Norte, Av. Sen. Salgado Filho, 1559 - Tirol, Natal - RN 59015-000 Brazil; joao.vilasboas@ifrn.edu.br

**Keywords:** Q-Learning, path planning, USV, ASV, autonomous sailboat, mobile robotics, green robotics

## Abstract

Path planning for sailboat robots is a challenging task particularly due to the kinematics and dynamics modelling of such kinds of wind propelled boats. The problem is divided into two layers. The first one is global where a general trajectory composed of waypoints is planned, which can be done automatically based on some variables such as weather conditions or defined by hand using some human–robot interface (a ground-station). In the second local layer, at execution time, the global route should be followed by making the sailboat proceed between each pair of consecutive waypoints. Our proposal in this paper is an algorithm for the global, path generation layer, which has been developed for the N-Boat (The Sailboat Robot project), in order to compute feasible sailing routes between a start and a target point while avoiding dangerous situations such as obstacles and borders. A reinforcement learning approach (Q-Learning) is used based on a reward matrix and a set of actions that changes according to wind directions to account for the dead zone, which is the region against the wind where the sailboat can not gain velocity. Our algorithm generates straight and zigzag paths accounting for wind direction. The path generated also guarantees the sailboat safety and robustness, enabling it to sail for long periods of time, depending only on the start and target points defined for this global planning. The result is the development of a complete path planner algorithm that, together with the local planner solved in previous work, can be used to allow the final developments of an N-Boat making it a fully autonomous sailboat.

## 1. Introduction

The number of researches working on the development of navigation methods for autonomous sailing has grown significantly. The interest in this research area is relatively new compared to general mobile robots, such as Unmanned Ground Vehicles (UGVs), Unmanned Aerial Vehicles (UAVs) and other types of motorized Unmanned Surface Vehicles (USVs). The main motivation for research into autonomous sailboats is the possibility of uninterrupted navigation by using the wind, a physically vast and free source of propulsion. This source is generally allied to solar panels and/or turbines that can be used to recharge internal batteries that feed the computers and actuators necessary for autonomy, based on green robotic concepts [1]. Besides, with more stability, a sailboat can carry a larger payload, which allows its use in a broad range of applications such as environmental and coastal monitoring [2] and other water quality related studies [3].

Nonetheless, since the propulsion source of a sailboat (wind) is usually complex and the sailboat is subject to other random forces such as currents and waves, its dynamics are highly non-linear, making its automatic control a non-trivial problem [4,5]. Particularly, by our experience, we noticed that path planning methods developed for general mobile robots will not work in the same way in this kind of autonomous vehicle. For example, the approach proposed by Chen et al. [6] based on the reinforcement learning (Q-Learning) [7] algorithm, originally developed and used on smart motorized ships does not work here in a straight implementation. It is difficult to use such an approach for sailboats, noting that motorized boats do not need to take into account wind direction. Fortunately, this restriction can be incorporated in the Q-Learning model by using a kind of punishment, still keeping the model as a Markovian decision process, thus enabling using the same approach for developing a path planning for a sailboat.

Thus, in the same fashion as Chen et al. [6], in this paper we propose to use Q-Learning for the global path planning of a sailboat robot using a different state space and set of actions. Our main goal is to find a feasible path that avoids obstacles while keeping the sailboat out of the dead zone. Q-Learning is a reinforcement learning approach that consists of rewarding or punishing agents according to actions that they perform in the world [7,8,9,10]. Furthermore, in this work we design a new cost function that rewards the agent and captures the desired behaviors representing the state space and set of action models both devised for modeling the behavior of the sailboat robot.

Our approach is different from the work of Chen et al. [6], which is not developed for sailboats, as we have to consider more situations. Our approach is able to find feasible trajectories even in situations against the wind while avoiding mapped obstacles. We validated our proposal with experiments in simulation which shows feasible sailboat trajectories for all situations, defined through the points generated by the algorithm.

The next section introduces the theoretical background showing concepts about path and trajectory planning, sailing and the Q-Learning algorithm. In Section 4, we devise the behavior model in which we generated the distance and reward matrices and the Q-Learning table according to wind and boat directions. Section 5 shows the behaviour of a sailboat in simulation while travelling through the path generated by the Q-Learning, in different scenarios with some discussion about them and, finally, in Section 6, we trace the new findings based on the algorithm results and limitations, with the conclusion about this work.

## 2. Theoretical Background

In order to better understand the concepts behind the proposed approach, a brief theory recapitulation about path and trajectory planning, sailboat navigation and reinforcement learning, especially the Q-Learning, including some terminology, are introduced in this section.

### 2.1. Path and Trajectory Planning

Path planning and trajectory planning are assuming an increasing importance in robotics. Path planning is simply the calculation of a path free of collision between a starting point and a goal in order for a robot to travel in an environment possibly with obstacles [11,12]. The solution to this problem plays a central role in the design of an autonomous robot navigation system. To complete the navigation task, the algorithms read a map of the environment or workspace and subsequently try to create free paths for the robot to traverse, avoiding objects and obstacles. There are several path planning algorithms such as road map, optimal search, heuristic algorithms, bug algorithms, and randomized algorithms. On its side, solving the trajectory planning problem means generating the reference inputs for the control system of the robot, so as to ensure that the desired motion is performed [13]. Usually, the algorithm employed for trajectory planning takes as inputs the path generated by the path planner, and the kinematic and dynamic constraints of the robot. The output of the trajectory planning module is given by the trajectory of the joints, or of the end-effector, in the form of a sequence of values of position, velocity and acceleration. The algorithms for trajectory planning are usually named by the function that is optimized, such as minimum time, minimum energy, and minimum jerk [13]. Hence, path planning generates a geometric path from an initial to a final point, passing (or not) through predefined via-points, either in the joint space or in the operating space of the robot, while trajectory planning algorithms take a given geometric path and endow it with the time information. In this paper, we deal with path planning.

A brief overview on autonomous mobile robot path planning focusing on traditional algorithms that produce optimal paths for a robot to navigate in an environment is presented by Sariff [11]. According to the author, the optimal path can be as short, smooth, and robust for autonomous robot navigation and simultaneously it has been proven that appropriate algorithms can run fast enough to be used practically without time-consuming problems. A more critical review on path planning techniques is presented in the work of Katrakazas [14], along with the associated discussions on their constraints and limitations, which seek to assist researchers in accelerating development in the emerging field of autonomous vehicle research. Several approaches exist to this problem in robotics [13,15,16,17]. In general, path planning algorithms use several methods for representing the environment, which can be as a configuration space or a graph presentations. Path planning algorithms are usually divided according to three methodologies, which are used to generate the geometric path, namely, roadmap techniques, cell decomposition algorithms, and artificial potential methods [13].

In this paper, we will focus on path planning, however, not in the strength and weakness of general path planning algorithms used by previous and current researchers, but in the development of high-level path planning for a specific type of robot, a sailboat, coming up with a particular algorithm that is effective and has shown to be robust. Nonetheless, we notice that sailboat navigation is a problem that has to be approached differently from any methodology that we found up to date in the literature for mobile robots as it has to consider the wind direction besides waves, which is not taken into account in the above approaches to general mobile robot platforms, including motorized USVs. These traditional algorithms do not work in the sailboat path planning due to the existence of directions in which the sailboat can not go forward besides not being occupied by any obstacles. So a different type of algorithm has to be devised for this particular problem, using Q-Learning, which will be treated further.

### 2.2. Sailing Theory

The main parts of a sailboat are, basically, the rudder, sail, hull and keel, as shown in Figure 1a. The sail and rudder are the core equipment, which are used for the control of the sailboat. The rudder’s main purpose is to change heading and the sail’s main purpose is to gather the wind force in order to give speed to the sailboat. The role of the keel is to help with roll stability, decreasing lateral drift and making the sailboat perform like a pop-up dummy on the water surface. In a sailboat, the keel has in general a heavy weight material on its bottom tip to lower the center of mass of the boat and avoid capsizing. The hull is the body of the sailboat. Its front is called bow and the back is often called stern. The left side of the sailboat is called port and the right side is called starboard. Some physical greatness of importance appear here as the true wind vector, which is shown in the upper graphic of Figure 1b. It is defined in the n-frame and $\gamma_{tw}$ is the direction of the true wind vector. There is also the apparent wind, which is the wind measured in the sailboat Cartesian origin. Easing its understanding, it is the wind felt by the crew of the sailboat in a passenger sailboat version. Notice that it is given as the vector summation of the velocity of the boat and the wind direction.

The possible set of sailboat maneuvers is shown in Figure 2a, noticing that during sailing the sailboat can basically be found in three situations: upwind, downwind and beam reaching. In downwind sailing, the wind hits the sailboat from the stern so a sailor just needs to open the sail to gather wind and gain speed. In the beam reaching situation, the wind hits the sailboat from the sides, with an angle of 90 degrees. In upwind situations, there is a range of directions where it is still possible to gain some speed. Actually, certain types of rigid sails hit their higher performance in this situation. In this case, as the sailboat speed increases, the apparent wind speed also increases, so it is possible to sail faster than the wind. Nonetheless, the region directly upwind, of approximately 30° from one side and the other, is called the dead zone or no-sail zone. In this region the sailboat can not gain forward speed due to the fact that the angle of the wind on the sail does not allow propulsion. The minimum angle of sail, i.e., the closest angle that a sailboat can travel against the wind can be found through wind tunnel experiments, measuring the force generated by the sail as it goes closer to the wind direction. Alternatively this can also be determined during field trials.

Beating is a zigzag maneuver that a sailor performs in order to reach points that are in the dead zone or straight against the wind. This maneuver consists of a number of tacks in which the bow crosses the wind and the wind changes from port to starboard (or from starboard to port). In this way, the sailboat is able to be kept outside the dead zone while advancing towards the target. Figure 2b shows an example of a tack. In situation 1, the sailboat is preparing to tack and the wind comes from starboard. In situation 2, the sailboat is performing the tack and crossing the wind. The sailboat uses its inertia to keep some velocity across the wind direction as it does not have propulsion during this state, so a minimum speed is necessary to perform this maneuver. In situation 3, the sailboat finishes the tack, and is starting to regain speed with the wind coming from port.

### 2.3. Q-Learning

Q-Learning is a class of reinforcement learning algorithm based on the Markov Chain theory, as illustrated in Figure 3. Basically, the agent is at an initial state and should apply an action $\left(A_{t}\right)$ in the world. After executing this action, the world returns to the agent its next state $\left(S_{t}\right)$ and a reward $\left(R_{t}\right)$ is given according to some cost function. In this scenario, the problem that the agent tries to solve is to find a sequence of actions, called a policy, that takes the agent from a current state to a desired (final) state while maximizing the immediate rewards (exploitation) and the sum of rewards to the target (exploration).

The Markov property states that the current state the agent is using to make its decisions has the sufficient statistics of the history of that agent. Mathematically, a state $S_{t}$ is a Markov’s state if and only if
(1)$$p\left(s_{t+1}|s_{t},a_{t}\right)=p\left(s_{t+1}|h_{t},at\right)$$
where *t* is the time step, $S_{t}$ is the state in time *t*, $a_{t}$ is the action in time *t* and $h_{t}$ is the history in time *t*.

In this sense, the Q-Learning algorithm is based on the Markov Chain added the rewards, actions and the policy concepts to the mathematical formulation, which is known as a Markov Decision Process [18]. In order to better understand this formulation, some concepts are presented below.

The return $G_{t}$ is the discounted sum of rewards from time step to the number of time steps in each episode as defined by Equation (Equation 2)
(2)$$G_{t}=r_{t}+\gamma r_{t+1}+\gamma^{2}r_{t+2}+\gamma^{3}r_{t+3}+\cdots$$

The discount factor (γ) is a term that is mathematically convenient since it avoids a large number of return values. When γ=0, $G_{t}$ only takes into account immediate rewards. On the other hand, when γ=1, future rewards are as beneficial as immediate ones. The value of γ is always set between 0 and 1. In our case, the state value function that will be updated is given by
(3)$$V\left(s\right)=E\left[G_{t}|s_{t}=s\right]=E\left[r_{t}+\gamma r_{t+1}+\gamma^{2}r_{t+2}+\gamma^{3}r_{t+3}+\cdots |s_{t}=s\right]$$

By rearranging the terms of Equation (Equation 3), the state value function is equal to the immediate return of this state (R(s)) plus the discount factor multiplied by the sum of probabilities of all states due to the previous state, multiplied by the state value function of the next state. Mathematically, this can be written as
(4)$$V\left(s\right)=R\left(s\right)+\gamma \sum_{s^{\prime }\in s}P\left(s^{\prime }|s\right)V\left(s^{\prime }\right)$$

Notice that Equation (Equation 4) regards only the rewards. By adding the influences of actions and establishing a policy, the complete modeling of the agent behavior can be set as
S: a finite set of Markov states (s∈S);A: a finite set of actions (a∈A);p: the model for each action, that specifies $p(s_{t+1}=s^{\prime }|s_{t}=s,a_{t}=a)$;R: a reward function given by $R\left(s_{t}=s,a_{t}=a\right)=E\left[r_{t}|s_{t}=s,a_{t}=a\right]$;γ: a discount factor, where γ∈[0,1];and π: a policy given by $\pi \left(a|s\right)=p\left(a_{t}=a|s_{t}=s\right)$

The new *R* and *P* functions are dependent on the policy and will be, respectively
(5)$$R^{\pi }\left(s\right)=\sum_{a\in A}\pi \left(a|s\right)R\left(s,a\right)$$
(6)$$P^{\pi }\left(s^{\prime }|s\right)=\sum_{a\in A}\pi \left(a|s\right)P\left(s^{\prime }|s,a\right)$$

The state value function is given by
(7)$$V_{k}^{\pi }\left(s\right)=r\left(s,\pi \left(s\right)\right)+\gamma \sum_{s^{\prime }\in s}p\left(s^{\prime }|s,\pi \left(s\right)\right){V^{\pi }}_{k-1}\left(s^{\prime }\right)$$

### 2.4. Q-Learning Values Update

Equation (Equation 7) is known as a Bellman backup and it is the basis of the Q-Learning algorithm. The algorithm idea is to find the optimal policy that specifies the best sequence of actions. For this, the goal is to maximize the return value ($G_{t}$) defined in Equation (Equation 2). Thus, a function Q(s,a) is defined that is the expected value of discounted rewards if performing action *a* from state *s*. By following an optimal policy for all states and defining
(8)$$V\left(s\right)=arg\max_{a}Q\left(s,a\right)$$
the function Q is nothing more than a state value function as show in Equation (Equation 7). It takes into account that its variable values are updated during the learning process without it being necessary to use a probabilities matrix for this. This update can be done using deterministic and probabilistic models. For a deterministic scenario, it is just a matter of performing action *a* in state *s*, get the reward *r* and update the value of Q(s,a) by adding it with the previous value. Mathematically, this can de written as
(9)$$Q\left(s_{t},a_{t}\right)\leftarrow r+\gamma \max_{a_{t+1}}Q\left(s_{t+1},at+1\right)$$

On the other hand, the probabilistic updating rules can be computed by way of using Equation (Equation 10):(10)$$Q\left(s_{t},a_{t}\right)\leftarrow \left(1-\alpha \right)Q\left(s_{t},a_{t}\right)+\alpha \left[r+\gamma \max_{a_{t+1}}Q\left(s_{t+1},a_{t+1}\right)\right]$$
where α is the learning rate.

The convergence of the Q-Learning Algorithm has been proven in the work of Melo [19] and we recommend reading it to better understand the concepts of its mathematical formulation.

Three more definitions are important in order to understand the Q-Learning approach used in this work. The first one is the *horizon* that corresponds to the maximum number of iterations that will be performed by the algorithm in the learning processing. The second one is related to the *random explorer* value. This value is useful because, in some cases, the algorithm uses a random action instead of the maximum value of the Q-Table. This avoids the algorithm entering in a loop between actions during the learning steps. Finally, the parameter used in this work to evaluate the learning process is the temporal differential (TD) [8] accumulated by the steps. This value is computed directly from the Q-Learning Equation (Equation 10). By rewriting this equation and isolating the part of it that is multiplied by the learning rate, we obtain the value of TD as shown in Equation (Equation 11). The sum of this value in each step is the accumulated error (Equation (Equation 12)).
(11)$$TD\left(t\right)\leftarrow r+\gamma \max_{a_{t+1}}Q\left(s_{t+1},a_{t+1}\right)-Q\left(s_{t},a_{t}\right)$$
(12)$$TD_{\mathrm{acc}}=\sum_{t=0}^{t_{\max}}TD\left(t\right)$$

### 2.5. N-boat Control Architecture

The N-boat control architecture (Figure 4) is organized as follows. The low level is composed of both heading and propulsion control, which work independently. The propulsion control changes the sail’s position so the sailboat has forward speed according to current wind speed and direction. The heading control changes the rudder’s position so the sailboat can follow the line that connect the last and the coming waypoints. At the higher level, we have the local and global path planning modules working. The local path planning uses information about surface and underwater obstacles to find alternative paths to avoid them. At the highest level, we have the global path planning, which uses information about the climate and terrain to find feasible routes for the N-boat. As it will be shown latter, this is where our proposal resides, i.e., to find these routes using the Q-Learning algorithm.

The basic processing flow of the N-boat Q-Learning algorithm is to create a simplified map of the terrain in order to get a discrete description of the environment. Next, the wind map is looked up to get its direction and to remove the actions that put the N-boat inside the dead zone, so a safe path can be learned. Finally, the algorithm is trained so a safe, feasible path is found and passed on to the other lower layers.

## 3. Related Works on Sailboat Path Planning

A common guidance for the software architecture of mobile robots is the separation of the path planning in local/short-term and global/long-term. Stelzer [20] defines long-term path planning for autonomous sailboat as the use of a global weather map (wind, currents and/or other environmental forces) to find optimal routes to the target while short-term path planning uses only local and instantaneous information about the weather. Local path planning is usually reactive, to allow avoidance of non-static and non-mapped obstacles, while long-term path planning is deliberative in the sense that it uses a global map and occupancy-grid-based algorithms [21] to find a particular path to the objective. Determining the global path (also called weather routing) for a sailboat is of extreme importance in long range missions since guidance based only on local and instantaneous wind information can lead to undesired and potentially catastrophic situations.

A usual cost to optimize in long-term path planning for motorized mobile robots is the distance to target, so a methods can try to find the shortest path. Mapping approaches based on occupancy grids are the most common although other approaches for calculating the maps are used. This is not enough for sailboats since the path of the shortest distance is not guaranteed to be the fastest one and it can even be unfeasible due to the sailboat kinematics. When the sailboat has to move with the same wind direction it can travel without restrictions. On the other hand, a problem appears when the sailboat has to travel against the wind, due to the existence of a region called the dead zone (or no-sail zone), where the sail is unable to generate propulsion. So a good starting point for path planning methods for sailboats is trying to generate, at least, feasible paths.

Compared to more traditional motorized research mobile vehicles, such as UAVs, UGVs and even motorized USVs, there is a limited amount of studies on autonomous sailboat navigation. The current state-of-the-art in sailboat navigation can be classified as deterministic and non-deterministic or stochastic path-planning. In the work of Baker et al. [22], a global map is used with the Dijkstra algorithm to find the shortest path to the target without taking wind and current information into account. Saoud et al. [23] developed a PRM–Dijkstra (PRM: Probabilistic Road Map) algorithm that basically divides the path into segments and uses the expected time to travel a segment according to information about wind and the sailboat polar diagram as a cost function. Their method also provides avoidance of mapped obstacles. Langbein, Stelzer and Frühwirth [24] use an $A^{*}$ to find optimal routes in dynamically adapted weather maps, while Cabrera et al. [25] use a Nelder-Mead simplex algorithm on weather maps to find optimal routes that minimizes the time to the target. This last method also accounts for obstacles in the map.

A class of more recent researches in autonomous sailboat path-planning is heading towards meta-heuristic and adaptive/intelligent methods. In the work of Du et al. [26], a method called three-dimensional dynamic programming (3DDP) is used to generate multiple routes between two points in a weather map. Those routes are optimized for time to destination and wind direction and the best path can be chosen according to other factors, such as safety and energy consumption. In the work of Wang, Yao, and Dou [27] a comparison of PSO (Particle–Swarm Optimization), GWO (Grey–Wolf Optimization) and a modified GWO (IGWO (Instinctively Grey–Wolf Optimization)) is presented. These methods are used to find energy-minimum and shortest paths while avoiding mapped obstacles. The case study is a sailboat robot but all experiments were performed in downwind situations, so performance in upwind situations is uncertain.

Q-Learning has also been used successfully in other robot platforms such as UGV and AUV. Jaradat et al. [28], Khriji et al. [29] and Jiang et al. [30] used classical Q-Learning to find obstacle-free paths for differential UGVs in dynamic environments. To increase the convergence speed of training the Q-Learning, the above studies reduce the number of possible actions of their robots. Jaradat and Jiang used only eight actions to represent the robot movement while Khriji used 14 actions. Cui et al. [31] and Zhang et al. [32] used modified Q-Learning approaches to solve the problem of path-planning and obstacle avoidance for UAVs. The idea is that since a UAV acts in a dynamic and uncertain environment subjected to wind forces, state estimation and actions are usually prone to errors. Hence, the navigation problem can be modelled as a partially observable Markov decision problem, and Q-Learning is a valid solution for path-planning in unknown environments.

The approach proposed by Chen et al. [6], which is based on the Q-Learning algorithm [7] and was originally developed to be used on a fleet of smart motorized ships is the closest research to ours and has also motivated our proposal. The authors argue that the essence of path planning based on the Q-Learning system is that agents can independently find the most effective path by enumerating all possible solutions, which might be closer to human manipulating intelligence. Further, the Q-Learning can find good solutions even in an uncertain and chaotic aquatic environment. The only prerequisite is to build a computing environment which is consistent with or close to the real world. Nonetheless, note that motorized boats do not take into account wind direction, so it is difficult to use such an approach as is for sailboats. It needs further developments to be adopted here, as will be described next.

## 4. Path Planning Implementation

The first step is the generation of a map where the Q-Learning approach will be applied to define the trajectory. Notice that, in the real world, there are some regions, other than the dead zone, where the sailboat can not pass, such as obstacles and the borders of the water body. These regions are joined together as possible generating blocks. These blocks should be mapped so the Q-Learning algorithm can generate feasible paths avoiding them. After mapping, these blocks have a penalty associated to them, which indicates that it is not easy for the sailboat to go through that region and so the algorithm needs to search for an alternative path.

To simplify the obstacles and borders matrix generation, we have created an algorithm using the OpenCV library [33] in order to identify the blocks on an image with known obstacles painted in red color. This is done by computing the width and height of the image, and dividing it by the number of squares on the reward matrix and looking for the central point of each square. Then mapping the blocks in a matrix N × N mapping the regions where the agent can not reach.

Figure 5 shows the construction process of the image in order to create the blocks matrix. Figure 5a is a piece of Bonfim lake taken from Google Maps marked with blue squares on the area of interest and red squares are placed on the borders and obstacle points. Figure 5b corresponds to the final image that will be the basis for the algorithm to generate the blocks matrix.

### 4.1. Generating the Reward Matrix

The reward matrix is obtained from another matrix called the distance matrix. The values of this last matrix correspond to the number of squares from any point on the map to the end point. Figure 6 exemplifies the distance and reward matrices, respectively. The yellow square is defined as the end point and the distance matrix is calculated from them (Figure 6a), where the side squares have a distance of 1 and diagonal squares have a distance of 2. The values of any other square corresponds to the number of remaining squares to reach the final point. On the other hand, the values of the matrix in Figure 6b are the rewards to get from any place on the map table to the end point. Notice that the reward matrix is obtained from the difference between each value of the distance matrix by the highest distance value.

Other aspects that should be taken into account are the obstacles and borders. For these cases, the reward is actually a penalty applied to the agent that allows it to know that a block should not be achieved. Generally, it is given a number much lower than that used for the rewards, like minus 1000 plus the negative of the highest reward, for example. Notice that this penalty can be added to the reward matrix after its generation as shown previously. This will not change the algorithm efficiency due to the necessary conditions: (1) there should be a possible path between the start and end points, and (2) the end point can not be placed in the same point, and should be different from obstacles and borders.

### 4.2. Q-Table Generation

Once the reward matrix has been generated, it is necessary to generate the Q-Table that will be used in the learning process. The table is composed of the actions on the rows and the states on the columns. The actions define which directions are allowed in that particular state.

Specifically for this application, as the wind direction must be taken into account, the definition of the actions is very important in order for the desired goal to be reachable. As mentioned above, there is a range of angles between the wind and the boat directions that is called the dead zone. In this zone, the boat is not able to go straight ahead because it is against the wind and, in order to achieve the final point in this situation, it is necessary to perform a zigzag trajectory. This is the beating, where the boat changes its orientation alternating i order to get out of the dead zone, thus being able to reach the goal point at the end.

Figure 7 shows all the possible actions used in this work related to the world (circle) and to the algorithm (table) without taking the wind into consideration. Notice that when the wind is taken into account there are some actions that should not be done because they might get the boat into the dead done. To illustrate this, Figure 8 represents some possible trajectories for different scenarios. The start point ($S_{p}$), the end point ($E_{p}$), and the wind direction are shown.

In the cases where the desired heading to get to the end point is outside the dead zone (Figure 8a), all actions available may be used in the Q-Table to generate the path. Otherwise, when it is necessary to perform a zigzag as shown in Figure 8b some actions have to be removed from the action pool and they can be chosen only the actions that are allowed by the policy of each scenario.

Figure 9 helps understanding how the approach is implemented. Figure 9a presents the relationship between the boat and wind directions to indicate whether the boat is inside the dead zone. If the dead zone is confirmed, then it is computed the range of directions that the sailboat cannot follow. Thus, it is mapped which actions corresponds to those angles as shown in Figure 9b.These actions are then removed from the pool of actions and the Q-Table is generated using the remaining ones. Notice that depending on the dead zone range multiple actions could no longer be feasible. For instance, in the situation exemplified in Figure 9 the dead zone range is 80°, which results in two actions that were removed from the action vector.

Mathematically formalizing this, the dead zone ($\Delta_{dz}$) is defined by:(13)$$\Delta_{dz}=[\gamma_{tw}-|\psi_{dz}|,\;\gamma_{tw}+|\psi_{dz}\left|\right]$$
where $\gamma_{tw}$ is the angle of the true wind, $\psi_{dz}$ is the minimum angle of sailing in upwind.

If ($\psi_{b}-\pi )\in \Delta_{dz}$, then
(14)$$A_{k}=\left\{\begin{array}{c} 0,\;k\in \Delta_{dz}\\ |k-\pi |,\;\;\;otherwise \end{array}\right.$$
where $A_{k}$ is a vector that contains the actions that will be used to generate the Q-Table, and
(15)$$k=n\frac{\pi }{4}\;\;n=\left[0,1,2,\ldots ,6,7\right]$$

The operation |k−π| is necessary because the actions are defined in the sailboat frame. Thus, the action vector is contrary to the wind direction.

## 5. Experiments and Results

In order to validate our proposal, we start with a set of experimental scenarios that were designed in order for the algorithm to be used to generate paths to go from a point (source) to another (target), in a lake-like environment. These experimental scenarios and situations use similar maps as shown above in Figure 5a. Our usv_sim simulator (https://github.com/disaster-robotics-proalertas/usv_sim_lsa) [34] is used in this work to perform the trajectories based on the paths resulting from the experiments. These multiple scenarios were created for the tests, basically, by changing the initial and end points and also the wind direction. In scenarios 1 and 2, the wind direction is fixed, and the start and end points are then swapped in each of them to simulate both upwind and downwind situations. The first scenario is free of obstacles while the second has some obstacles that were added near the border in order to force the algorithm to change the generated path. The resulting block matrices for each scenario are as shown in Figure 10. In scenario 3, the start and end points are fixed and the wind direction is changed in order to observe if the Q-Learning would still find feasible paths. Additionally, we also devised other scenarios for comparison with the A* algorithm. Besides, this algorithm does not apply to sailboats (mainly in upwind situation).

In each scenario, the wind vector is considered to be uniform throughout the whole map, with the same direction, which may vary from one scenario to another, and with a wind speed of 2.2 m/s. The parameters used on the training are shown in the Table 1. These values were found empirically by applying a fine tuning on the error and the maximum rewards achieved in each training.

### 5.1. Scenario 1 - Path without Obstacles

In this scenario, there are only the border blocks without obstacles. Figure 11 shows the experiments and resulting sailboat paths for this scenario. Basically, we devised a mission where the boat goes from the start point to the end point (left of Figure 11) and then returns to the start point (right of Figure 11). The white marks are the waypoints generated by the algorithm. The green square is the start point whereas the gray one is the target point. The red lines are further trajectories performed by the sailboat in the simulator, using the path given by the path planning.

Note that in Figure 11a, the direction to the end point is downwind. For this reason, all actions, as shown above, are made available to the algorithm and the path can be done by the shortest distance, just following the diagonal path. In this case, the Q-Learning has been performed in such a way that is similar to other traditional algorithms for motorized boats. For example the A* (that will be shown below for comparison) would produce a similar path, besides being more complex to implement. On the other hand, in Figure 11b the direction to the end point is upwind and the A* does not work in this case. In this later scenario, our approach makes the boat ending up with a beating maneuver, performing a zigzag trajectory in order to reach the endpoint. The first experiments depicted in Figure 11 show that the paths generated by Q-Learning could be successfully used by the sailboat to generate its low-level trajectory and follow it reaching the target, both in downwind and upwind situations, and for scenarios with no obstacles. It can also be noticed that this result would be similar if there is no upwind situation by using other traditional approaches even not for sailboats. Nonetheless, notice that these strategies would not work in general situations mainly of upwind. In fact, we found no strategy that can be easily implemented, up to date, that can deal with upwind situations, so this is the major contribution of our work.

### 5.2. Scenario 2 - Path with Obstacles

A second scenario with the same wind parameters as scenario 1, but with the blocks map as shown in Figure 10b, is devised. Figure 12 shows the generated trajectories with the planned paths.

As in the previous experiments, Figure 12a shows the movement of the sailboat following the trajectory in downwind situations. In this case, since there is an obstacle in the middle of the diagonal line, the algorithm generates a detour avoiding it, and reaches the end point from another direction. In Figure 12b, the sailboat has to travel upwind and a zigzag trajectory was generated following a different path than the previous one. Notice that there is not an allowed action for the Q-Learning, which is the Down–Left one, during this experiment since the wind direction is the same. However, the path could be created without considering this action that was excluded by the algorithm. Nonetheless, the experiment shown in Figure 12 demonstrates that the paths generated by Q-Learning could be successfully used by the sailboat to generate a low-level trajectory and following it reaching the target while avoiding obstacles along the course, both in upwind and downwind situations.

### 5.3. Scenario 3: Changes in the Wind Direction

To evaluate the robustness of the method, we created another scenario with three situations having the same start and end points but with changes in the wind direction in order to verify the possibility of this approach to calculate feasible paths. This can be seen in Figure 13. In the first case (Figure 13a), the sailboat is moving with the same wind direction, so the feasible path that the Q-Learning generates is also the path of shortest distance. In Figure 13b, the end point is against the wind and the Q-Learning generates a feasible path with two straight segments. In the third case (Figure 13c), the end point is also against the wind but with a sharper angle, so the Q-Learning generated a feasible path by adding multiple zigzags. It can be seen in the figure that, by following this path, the sailboat devised a trajectory that reached the target.

In addition to calculating feasible paths, the experiments shown in Figure 13 demonstrate the algorithm’s ability to adapt to different wind directions. In Figure 13a, the boat is not inside the dead zone and its movement is in the same direction of the wind. The generated path is initially a diagonal line and then turns to the up direction until the end point is reached. The other two figures (b and c) show the situation where the sailboat is in the dead zone. However, in Figure 13b, only the up-left direction is blocked, and in Figure 13c, both up–left and up are not allowed. As noticed by an expert in sailing within our group, the trajectory followed in Figure 13c is not the best trajectory for this case, but it is a feasible trajectory and the most important is that it solves the problem for the proposed scenario.

### 5.4. Q-Learning Convergence and Performance

Related to Q-Learning convergence, Figure 14 shows the temporal differential (TD) accumulated error for the first two scenarios described above (Figure 11 and Figure 12), which were computed by using Equation (Equation 11). We noticed that the learning cost is proportional to the scenario complexity. The scenario without obstacles and with the sailboat direction being the same as the wind (blue line) converges more quickly. On the other hand, the scenario with obstacles and sailboat direction against the wind (inside the dead zone, the red line) takes longer to achieve the convergence. These results show that the wind direction is an important parameter in path planning for sailboats because it directly changes the computational cost in the learning process. This is also true for traditional strategies as will be discussed below.

In relation to performance and starting with time spent, for the tests performed in this paper, the execution time of the training was about 1 minute and 8 seconds for a horizon equal to 10,000 steps of training, running on a computer with an ${\mathrm{Intel}}^{®}$${\mathrm{Core}}^{\mathrm{TM}}$ i7-8565 1.8 GHz CPU and 16 GB RAM memory. This value changes depending on the scenario and number of steps (actions). For instance, the scenario without obstacles and the sail boat with same wind direction, the training time was 6 seconds with 1000 steps. This information is important because in a real environment, where there are other kinds of obstacles (other boats in movement, for example) and more complex situations like severe water flow, it would be necessary to recompute the path in real time in order to allow the sailboat to make decisions and eventually re-plan the route, creating alternative paths to avoid obstacles that this will show up only during the mission execution. However, notice that situations like these are generally treated with other low-level approaches that work in parallel with this high level path planning layer. In fact, we see this as an advantage of this algorithm that can even be used anytime to recompute a path from the actual sailboat position to the target without much trouble.

Another issue that affects the algorithm efficiency is about the quantitation of the grid. Since in this approach, the real world is mapped on a matrix with N × N positions, there might be some loss of reachable points. Nonetheless, the higher the discretization is, more points in the environment can be achieved, but this significantly increases the processing of the training step. On the other hand, the lower this discretization value is, the faster the generation of the path, but fewer points in the real world can be reached with precision. To this end, the right measurement between the number of matrix squares and the number of points available depends on the mission details and this was determined empirically in this work.

### 5.5. Scenario 4: Comparison with A* and An Expert Sailor

We found no algorithms in the literature that work in a similar way for high-level path planning of sailboats. We devised a last set of experiments to provide some comparison of our approach mainly with the A* algorithm [24], which we implemented only for a downwind situation as this does not work in upwinds. We also show paths devised by human sailor specialists, that can be used at some degree for visualizing the results. They were asked to draw the possible paths that they would follow, without informing them of the results of our implementations (Q-Learning and A*). This gives an idea of how optimized the path generated by the Q-Learning and A* is. Actually, Figure 15 and Figure 16 show the three approaches for a similar scenario in which the sailboat is out of the dead zone. Figure 15a is for the A* algorithm that surprisingly generated (visually) the same path as the Q-Learning algorithm (Figure 16a). This means that our approach is better than the A* since it also provides solutions for upwind situations, in which the A* can not be applied.

In Figure 15b, we show the possible path generated by the sailor experts (both agreed on it) that were asked to take into account the wind direction and to provide the smallest and/or fastest path to the target. Notice that the path given by the sailors is different from both automatic approaches, even in this downwind situation. This is because they considered minimizing the number of jibes necessary on the path besides the wind direction. Each time the wind changes from starboard to port (or the opposite) a jibe is necessary, which makes the boat lose velocity.

Furthermore, Figure 15c shows the path provided by the sailors with the start and target changed around. When comparing this path to the one resulting from the Q-Learning shown in Figure 16b the paths are very similar. In this opposite situation, the A* is not able to find feasible paths. This is because it will eventually end up in an upwind situation which has to be considered, but the A* will ignore it, eventually directing the boat against the wind at some point. At this time, the sailboat will eventually start losing velocity until it completely stops.

### 5.6. Further Discussion of the Results

From the experiments, we notice that the application of reinforcement learning, especially the Q-Learning algorithm, to the generation of paths for an autonomous sailboat is interesting because the actions forbidden by the sailboat kinematic are easily translated into the Q-Table. The actions used to generate the Q-Table are, in our case, the possible directions to be followed by the sailboat taking into account the wind direction, obstacles and borders. In other words, these directions are directly managed by the algorithm in the form of actions. This complex behavior was achievable simply by allowing and/or removing actions from the matrix generation.

Lastly, the low-level control used for the sailboat in the experiments is the simplest control strategy for a sailboat to follow a path, comprised by a PID for heading control and the linear selection of sail angle based on the wind direction for propulsion control. This simple control was enough to follow the generated trajectory without the addition of more complex control strategies. This shows the robustness and feasibility of the route. On top of this low-level control strategy, the Q-Learning was capable of generating feasible paths for all the tested situations, including several ones against the wind.

Nonetheless, in the case where an obstacle was placed on the diagonal line generated in the first scenario (scenario 2), an avoidance of the obstacle in the path planned is provided, reaching the endpoint from another direction. Moreover, when the boat direction is contrary to the wind, the algorithm result is a zigzag trajectory also avoiding the obstacle. It is important to note that depending on the scenario, it is possible that the generated path could not be mapped onto a feasible trajectory. This is because the wind direction and obstacle avoidance conditions must be always satisfied, so obstacles can be put so close to each other in such a way that some requirement like this is not met. This possibility is also a problem for general path planning algorithms.

## 6. Conclusions

This work proposed the use of a Q-Learning approach for long-term path planning of autonomous sailboat. It has been shown that the idea of using such a reinforcement learning algorithm is interesting because it takes into account the kinematics of the sailboat, generating feasible paths while simultaneously avoiding mapped obstacles. The proposed strategy has been validated in a simulation environment. The experiments show that this approach results in correct trajectories that are generated for all possible situations (including upwind). With this, the map generation phase is automated, so it is easy to update maps, and information about currents, wind fields and additional obstacles that can be added during the run-time. Our approach is devised for situations that are similar to the works of Stateczny et al. [35,36] where the key aspect of autonomous navigation is the need to avoid collisions with other objects, including shore structures. In this case, we are also developing a Vision System based on the ZED camera that works well for distances closer than 20 m [37,38], besides using a LIDAR. So the N-Boat is able to automatically detect obstacles and perform suitable maneuvers in a similar manner as pointed in the above references. As pointed, this situation also arises in near-coastal areas, where shore structures like berths or moored vessels can be encountered.

We would like to highlight the importance of our algorithm to path planning in sailboat navigation, mainly in the upwind situations that really makes a difference, not allowing any motion if the boat eventually loses velocity and stops against wind situations. As pointed, the wind situation is complex in several regions of the world, even for example in the northeast of Brazil where we do have some regularity of strong wind coming from the sea. If an estimation of the wind conditions is given by some wind map, the algorithm can be used to find a path to the goal. Besides, during navigation, the real wind direction given by our embarked sensors can be used by the algorithm, even if it changes, which gives a new route from the current point to the target. With this developed technique, we are now able to complete the whole cycle, starting from high-level path planning to the low-level control of our sailboat.

We have shown that the traditional Q-Learning algorithm can also be applied with modifications for application to sailboat path planning, managing the allowed actions given wind direction, and decreasing the computational cost. As future work, we will try to improve the current implementation of Q-Learning in order to achieve better performance. For example, in order to smooth the generated trajectory, we can increase the number of Q-Table actions using Geometric Reinforcement Learning as proposed by Zhang et al. [32]. This allows the agent to search for the optimal next state in places other than its immediate neighbors. This approach will increase the computational cost, which is important to search for a balancing point between smoothing and cost. Another possible future work is to add the selection of actions according to the sail polar diagram in order to optimize the time to target. This allows better control of the number of tacks.

Another point for research is to find a trade off regarding the discretization of the map and processing time since this will increase the precision of the path while decreasing performance. Other strategies can also be devised for the sailboat robot to follow the generated path, with dynamica recalculation of the trajectory, if necessary, from each point. Approaches including visual attention features [39,40] and natural or artificial marks could also be devised for following a planned trajectory. Currently, these strategies are being implemented to work under ROS (Robotic Operating System) in order for them to be tested in the real N-Boat or sailboat robot (Figure 17) during field trials. Besides that, the model responsible for recomputing the path in real time due the change in the scenario will be added to the path planning layer, allowing the sailboat to travel a safe trajectory in all paths and situations. Deep reinforcement learning can be further studied here in order to find better and faster strategies that are optimized for other metrics besides feasibility, such as time to target and power consumption.

## Figures and Tables

**Figure 1 sensors-20-01550-f001:**
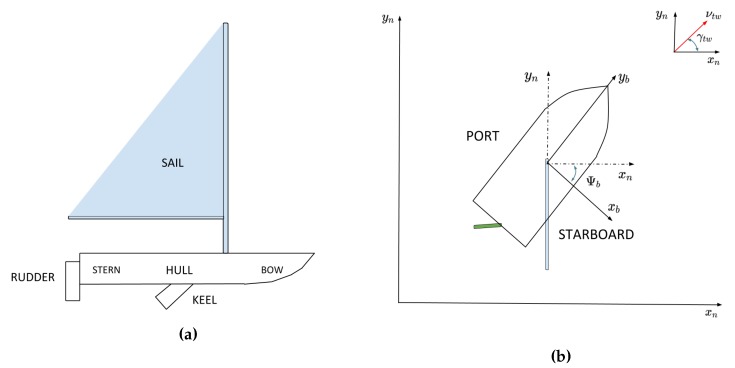
Main parts of a sailboat on (**a**). Frame of reference used in this paper on (**b**).

**Figure 2 sensors-20-01550-f002:**
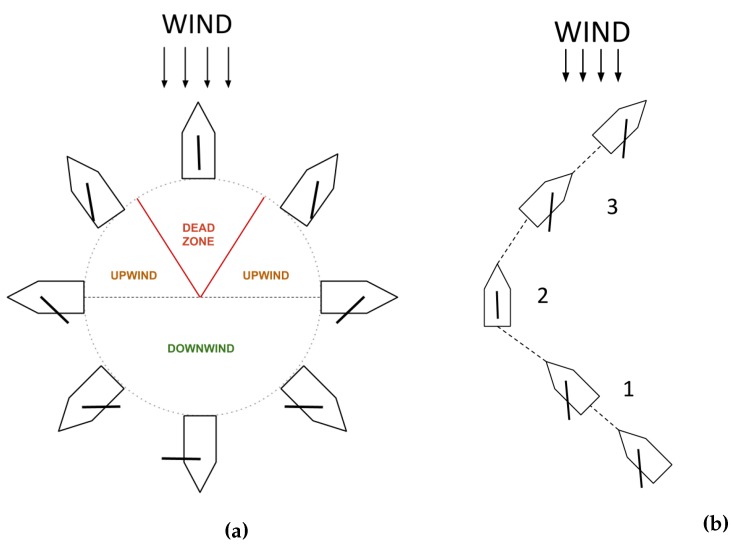
Sailing maneuvers on (**a**). Example of a sailing tack on (**b**).

**Figure 3 sensors-20-01550-f003:**
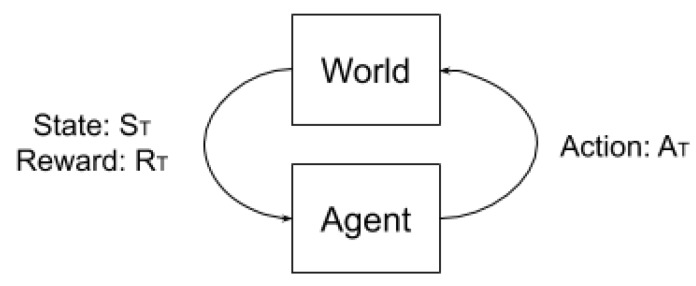
The Markov chain illustration. The agent acts in the world and receives back a state and a reward value. This return is used to update the learned values of the agent.

**Figure 4 sensors-20-01550-f004:**
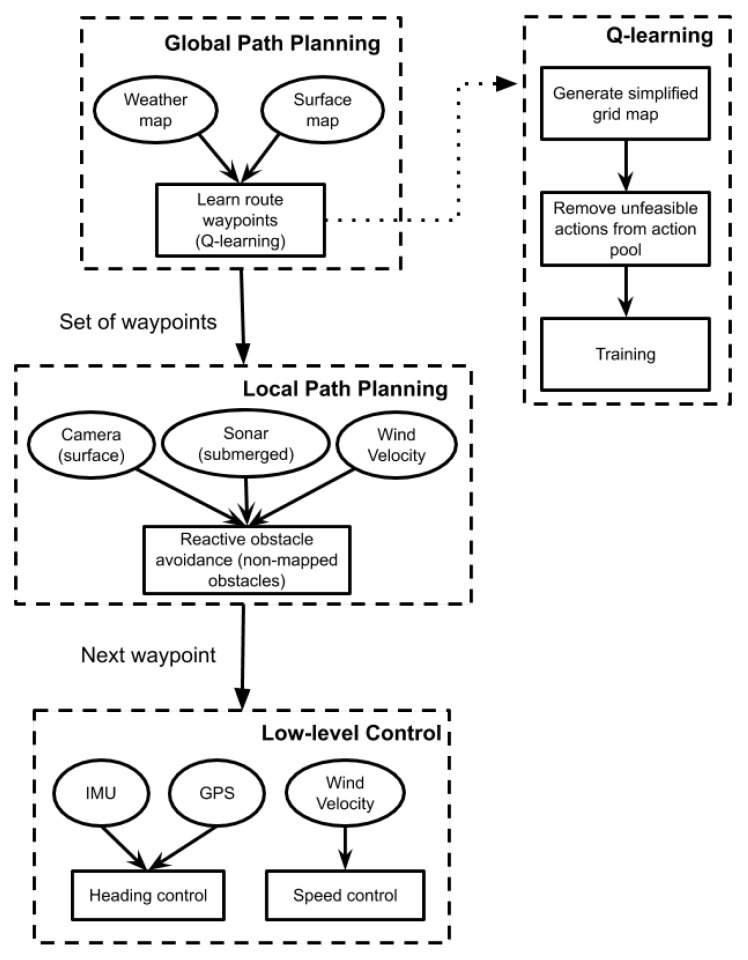
Data flow chart of N-boat sailboat.

**Figure 5 sensors-20-01550-f005:**
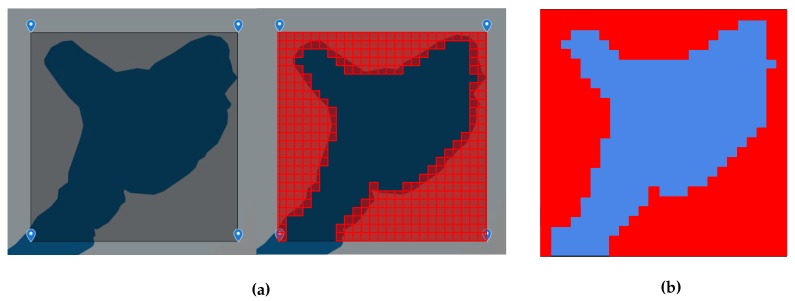
Creation of the image that will be used to generate the blocks matrix. In the most left image (**a**) we show part of Bonfim Lake (Natal, Brazil) inside the area of interest. Notice that in the center image, the red squares are placed on borders and obstacles. The right image (**b**) corresponds to the input of the algorithm to generate the block matrix.

**Figure 6 sensors-20-01550-f006:**
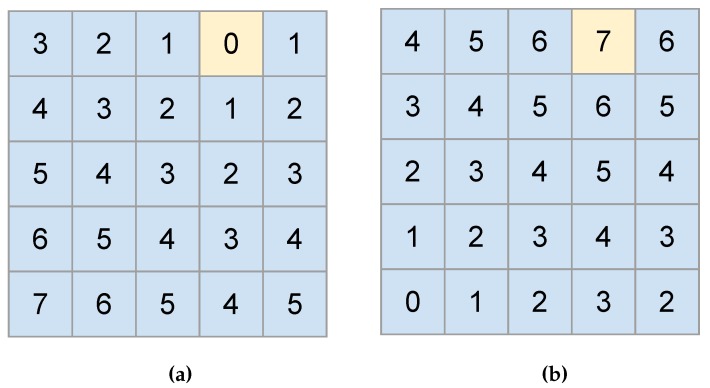
The matrices used on the generation of the Q-Table. In (**a**), we show the distance matrix with the end point painted in yellow color. In (**b**), we show the reward matrix that corresponds to the distance matrix minus the highest distance value, resulting in lower reward values to squares that are far from the end point, while the highest reward is on the end point.

**Figure 7 sensors-20-01550-f007:**
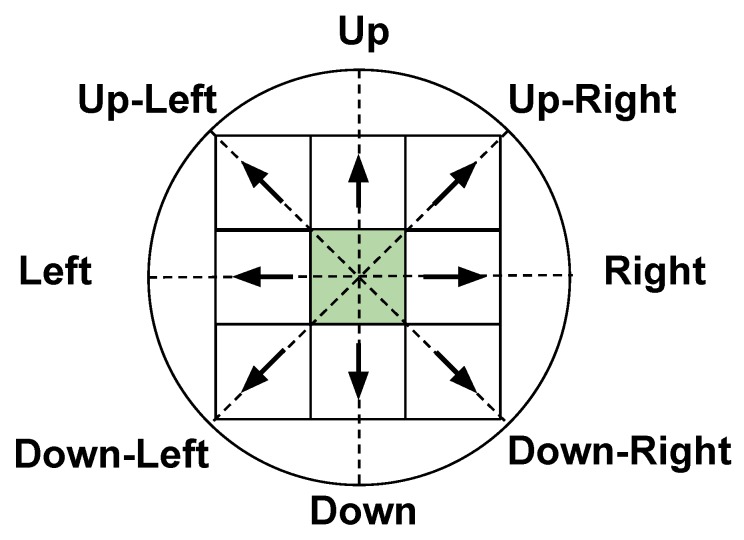
All actions available to generate the Q-Table. This figure is related the real world (the positions into the circle) to the simulate world used on the algorithm to create the trajectory (matrix).

**Figure 8 sensors-20-01550-f008:**
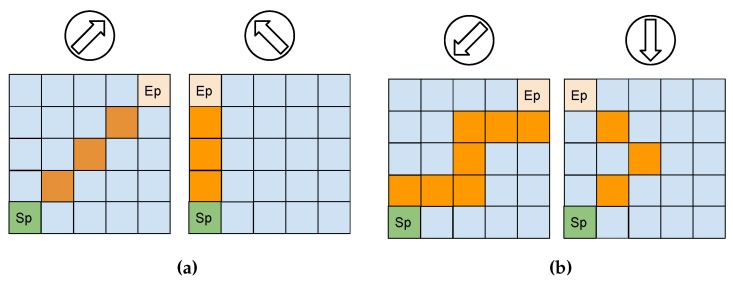
Four scenarios to exemplify possible paths from the start point (Sp) to the end point (Ep) taking account the wind directions. In (**a**) it is a downwind situation, so it is possible to reach the goal by following a straight path. On the other hand, in the two situations in (**b**) the wind and boat directions are contrary to each other, so it is necessary for the boat to perform a zigzag path to achieve the end point.

**Figure 9 sensors-20-01550-f009:**
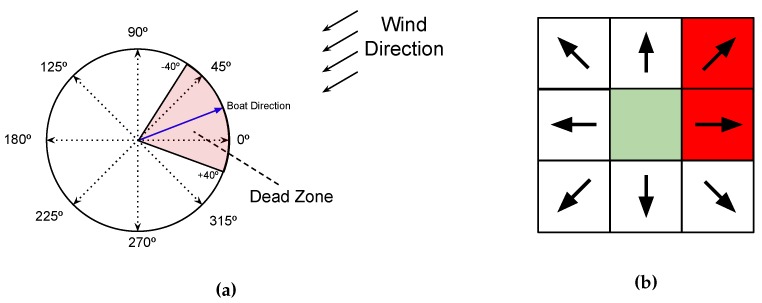
Relationship between the wind direction and the actions not allowed to generate the Q-Table. In (**a**), we show one example where the sailboat desired direction is contrary to the wind. In red color, the dead zone is presented with a range from +40 to −40 degrees taking account the boat direction. In (**b**), the actions forbidden are the red squares, while the other actions may be used in the Q-Table.

**Figure 10 sensors-20-01550-f010:**
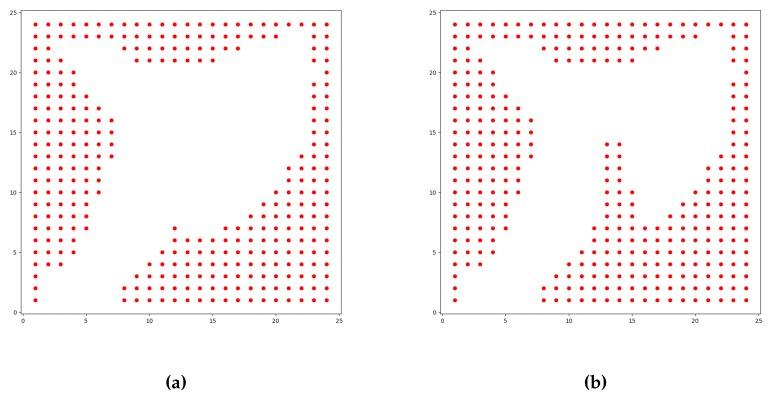
The block matrices resulting from the scenario shown in Figure 5b. In (**a**), we provide exactly the same map as already shown previously. In (**b**), there are some blocks added as obstacles to force the algorithm to change the trajectory generation.

**Figure 11 sensors-20-01550-f011:**
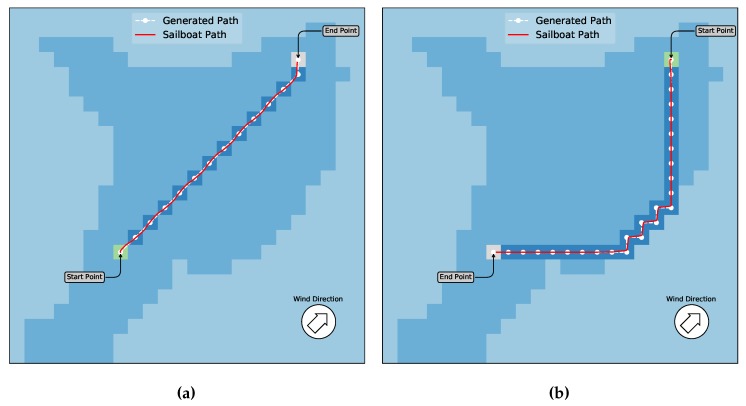
Scenarios without obstacles. In (**a**), the boat and wind directions are the same. Thus, the generated path can be done using all directions. The red line is the sailboat trajectory performed by the simulator with the path given by the algorithm. In (**b**), once the boat direction is contrary to the wind, a zigzag trajectory is generated. Note that the action down-left is not allowed in this case.

**Figure 12 sensors-20-01550-f012:**
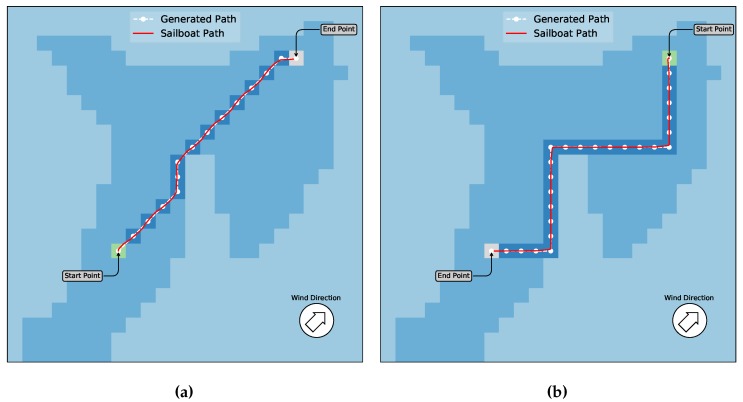
A similar scenario as in Figure 11b with obstacles added to the map. In (**a**), the boat and wind directions are the same. Thus, the path can be generated using all directions. The final path is changed if compared with the first case, in order to avoid the obstacles. The red line is the final sailboat trajectory performed on the simulator. In (**b**), once the boat direction is contrary to wind, a zigzag path is generated, with the action down-left not allowed in this case. Due to obstacles between the start and end points, the algorithm created a different path.

**Figure 13 sensors-20-01550-f013:**
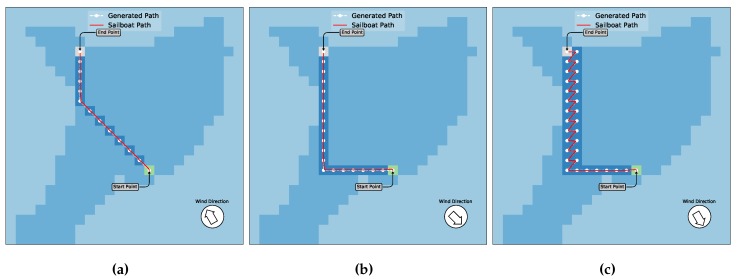
Three scenarios with the same start and end points and change in wind direction. The wind direction in Figure (**a**–**c**) are 315, 120 and 145 degrees, respectively.

**Figure 14 sensors-20-01550-f014:**
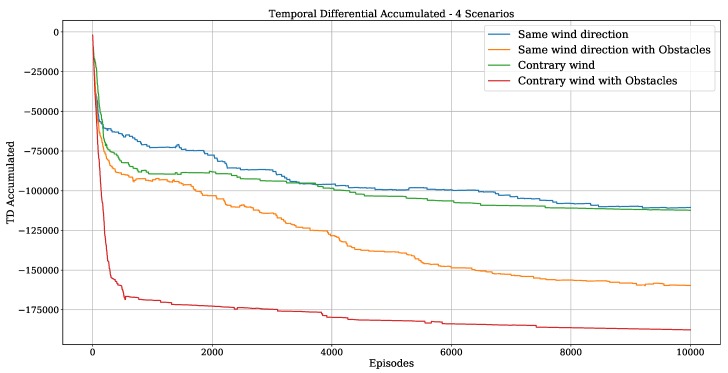
Accumulated Temporal differential error for scenarios Figure 11a,b and Figure 12a,b.

**Figure 15 sensors-20-01550-f015:**
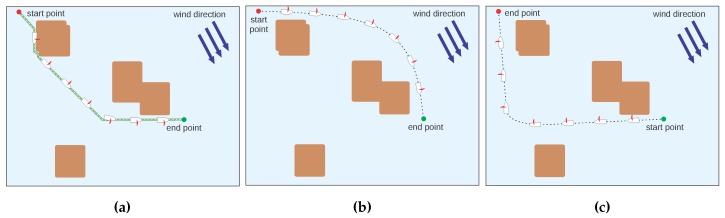
Path resulting from the A* algorithm and devised by sailor experts. In (**a**) the generated path by the A* algorithm, which is very similar to the Q-Learning resulting path shown in Figure 16a. Pictures (**b**,**c**) show the generated possible paths made by the sailor experts considering the wind in the same direction and versing the boat so that the start and end points change.

**Figure 16 sensors-20-01550-f016:**
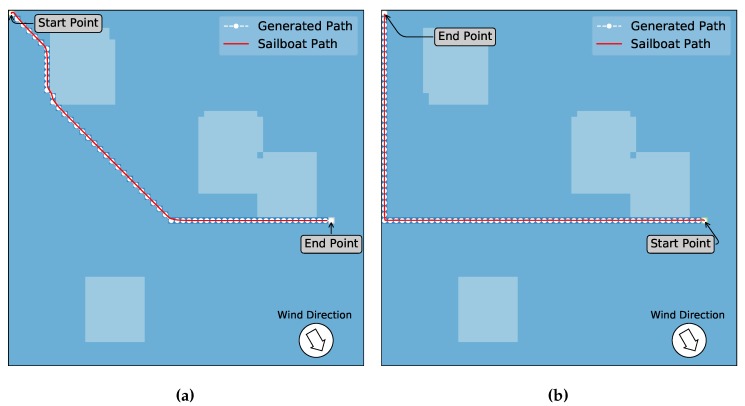
Q-Learning algorithm result to the scenarios shown in Figure 15 to comparison analyze. (**a**) presents the resulting path for target point in a downwind situation (sailboat and wind in the same direction). (**b**) shows the result for a situation where the sailboat should travel against the wind to reach the destination.

**Figure 17 sensors-20-01550-f017:**
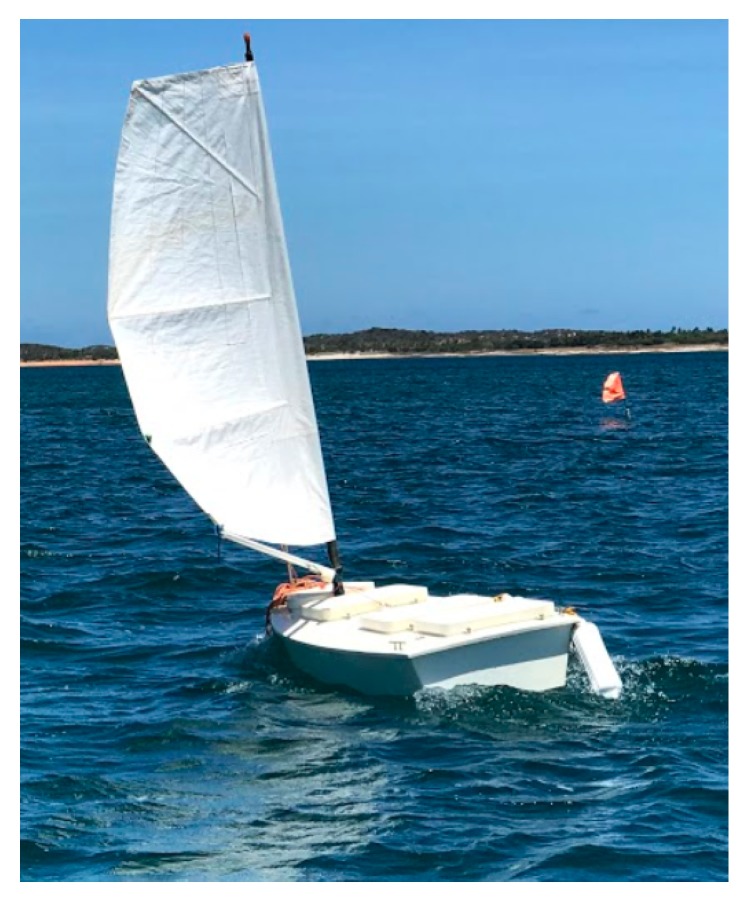
N-Boat the Sailboat Robot during the first field tests at the Bonfim Lake, Natal, Brazil.

**Table 1 sensors-20-01550-t001:** Parameters values used in each training.

Parameter	Scenario 1(a)	Scenario 1(b)	Scenario 2(a)	Scenario 2(b)	Scenario 3
Learning Rate	0.8	0.9	0.5	0.9	0.6
Discount Factor	0.7	0.7	0.3	0.7	0.4
Random Explorer	0.3	0.3	0.25	0.3	0.35
Horizon	10,000	10,000	10,000	10,000	10,000

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
