# Peer review of "High-Level Path Planning for an Autonomous Sailboat Robot Using Q-Learning"

_sensors, 2020, doi:10.3390/s20061550_

Round 1

Reviewer 1 Report

This paper presents the top (global) logical layer for the control of autonomous robotic sailboats, involving path planning based on statistical modeling and validated through simulations. The control system and the bottom (local) layer for the control of the autonomous sailboat are discussed by the authors in their 2016 Robotics and Sensors papers (references 3 & 4).

While the contribution of this paper in and of itself may be novel to the application in autonomous sailboats, the authors need to provide further background research and evidence supported by literature review of the state-of-the-art to solidify their claim. The background section is lacking in this regard and should be extended to compare similar path planning approaches in other arenas (ground and air) as well. 

While the simulation results show the results of several executions of the proposed algorithm, and attempt to briefly analyze the results via figure 13, the discussion of the relevance of the results is lacking in content. Again, the authors should compare the performance of their algorithm with the state-of-the-art, and expand the results section with more than the TDA graph, to better illustrate the performance and benefit of their proposed solution.

I would also recommend the manuscript be proof-read by a professional service, as some statements are lost in translation and awkwardly phrased.

Author Response

Please, see attached PDF file.

Reviewer 2 Report

The paper presents path finding algorithm, designed for sail boat. The all necessary aspects including sail theory was presented shortly, adequately and very clearly. Algorithm is very interesting, and application is now very attractive as for the ecology and energy saving. Paper has a very good quality. All sections are presented very clearly. I’m not native English speaker, but did not see any English  language issues.

Short questions for authors.

Is it possible to use this or based on this algorithm to planning path for hydrographic USV path planning, where available waters and restrictions are put similar to those putted in the paper. Particularly on lake applications and shore object detection like in:

Stateczny, A.; Kazimierski, W.; Burdziakowski, P.; Motyl, W.; Wisniewska, M. Shore Construction Detection by Automotive Radar for the Needs of Autonomous Surface Vehicle Navigation. ISPRS Int. J. Geo-Inf. 20198, 80.

And implemented on the system for this USV like:

Andrzej Stateczny, Paweł Burdziakowski, "Universal autonomous control and management system for multipurpose unmanned surface vessel", Polish Maritime Research, volume 26, issue 1(101), page 30-39, Jan 2019.

Author Response

Please, see attached PDF file.

Reviewer 3 Report

In this paper authors present high-level path planning for sailboat using q-learning. The language in the paper is satisfactory with few minor spelling and grammar mistakes and the paper itself is well structured.

The main contribution of the paper is in the expansion of the well-known q-learning path planning method in such a way to remove one or two directions (actions) corresponding to the dead zone of the sailboat caused by the wind.

The method is tested in simulations and have shown the promising results, but real life experiments would greatly improve the quality of the paper.

Even though the topic of the paper is interesting area of the autonomous sailboats, and the paper is relatively well written and presents some new methods and results, it not a journal material and is more suited for a conference publication.

Author Response

Please, see attached PDF file.

Round 2

Reviewer 1 Report

The authors addressed all issues raised in the first round of review, except for a crucial one. Again, the authors should expand the results section with more than a just the single analysis via the TDA graph, to better illustrate the performance and benefit of their proposed solution. While the few sample scenario runs illustrate their approach, the scientific rigor in the analysis of the performance of their algorithm is lacking. 

Author Response

Estimated Reviewer;

First we would like to thank you very much for the suggestions given by your revision and the insights that it has also given to us. We completely agree that it was missing some material in the experimental section, mainly a better explanation and discussion of the results shown in the figures. This is now done in this new version. Actually, we completely rewritten the Experiments and Results section by putting explanation and discussion just after each experiment and also by adding new experiments, including comparison to a traditional algorithm that we implemented in order to do some comparison (A*). As the A* can not be used in the general case of a sailboat path planning, we separate only a situation in which it could be applied. To our surprise, our Q-learning based approach performs similar as the A* algorithm in this situation (this is in the paper, in the results). Also, we know it is not that fair, however we also compared Q-learning with a human experts (we have two experienced sailors in our group, including a Master). Hopefully this new section meet now the standard that you required previously. Again, thank you so much for your revision.

Authors of "High-Level Path Planning for Autonomous Sailboat Robot Using Q-Learning".